# Uncertainty in Flood Mitigation Practices: Assessing the Economic Benefits of Property Acquisition and Elevation in Flood-Prone Communities

**William Mobley [1,\*]**, **Kayode O. Atoba [2]** and **Wesley E. Highfield [1]**

1   Department of Marine Sciences, Texas A&M Galveston, Galveston, TX 77554, USA; highfiew@tamug.edu
2   Department of Landscape Architecture and Urban Planning, Texas A&M University, Galveston, TX 77554, USA; kayodeatoba@tamu.edu
*   Correspondence: wmobley@tamu.edu

**Abstract:** Adopting effective flood mitigation practices for repetitive flood events in the United States continues to play a prominent role in preventing future damage and fostering resilience to residential flooding. Two common mitigation practices for reducing residential flood risk consist of raising an existing structure to or above base flood elevation (BFE) and acquiring chronically damaged properties in flood prone areas and restoring them back to serve their natural functions as green open spaces. However, due to data accuracy limitations, decision makers are faced with the challenge of identifying the financially optimal approach to implementing mitigation measures. We address this problem through the following research questions: What does the optimal allocation of flood mitigation resources look like under data uncertainty, and what are the optimal methods to combining mitigation measures with consideration for the best economic benefits? Using a robust decision making (RDM) approach, the effects of uncertainty in property values, construction and demolition costs, and policy implementation options such as structure selection and budget allocation were measured. Our results indicate that the amount budgeted for mitigation and how those funds are allocated directly influence the selection of the most economically viable mitigation practices. Our research also contributes to the growing need for evaluating specific flood mitigation strategies.

**Keywords:** uncertainty analysis; scenario planning; urban flooding; flood mitigation; property acquisition

## 1. Introduction

Effective flood mitigation practices continue to play an important role in preventing repetitive property damage in flood-prone communities, thereby fostering resilience to residential flooding. Two of the most common mitigation practices for reducing residential flood risk are elevating structures and acquiring damaged properties in flood prone areas. However, in practice, policies to implement these mitigation strategies are often reactionary and haphazard. Because of the complex nature of flooding problems, local decision makers are faced with the critical task of determining the most optimal ways to disburse recovery funds for these mitigation strategies while also maximizing economic benefits and flood risk reduction.

Elevating structures using fill, pilings, or other support structure to prevent inundation is one of the most common parcel-level flood mitigation methods [1]. This typically involves elevating a building to the 100-year level of inundation known as Base Flood Elevation (BFE) to prevent inundation from frequent flood events. Beyond BFE, some communities require freeboard to provide additional levels of protection from flood-induced inundation. Freeboard is calculated as the number of feet a building is raised above the BFE [2]. The height of freeboard and elevation vary for different

communities, but the Federal Emergency Management Agency (FEMA) recommends at least one foot of freeboard for structures in the 100-year floodplain. Other jurisdictions have stricter requirements requiring at least two feet of freeboard for new construction in the floodplain. Elevating structures can be costly and difficult depending on the existing foundation type. For example, "slab on grade" foundations are more expensive to elevate compared to structures on existing pile or pier and beam foundations. Funding programs from FEMA such as the Hazard Mitigation Grant Program (HMGP), Flood Mitigation Assistance Program (FMA), and Pre-Disaster Mitigation Grant Program all make provisions for funding property elevations. Local jurisdictions serve as subgrantees to grants received by the state through FEMA and are required to identify the spending priorities for the grants requested while also providing matching funds to execute the projects identified [3,4].

The second common mitigation method involves buying out flood-prone properties to restore the land back to its natural function as green open space. Since 1989, over 40,000 residential properties were acquired throughout the United States (US), primarily through FEMA programs [5]. The Federal Government passed the Disaster Mitigation Act after the catastrophic 1993 Midwest floods, which allowed for property acquisition and relocation of flooded properties and to help homeowners qualify for a buyout of their property at a pre-flood price [6]. HMGP is one of the most popular buyout programs and provides 75% of acquisition costs from the federal government while requiring local jurisdictions and agencies to match the remaining 25% [7,8]. Other programs include the department of Housing and Urban Development's (HUD) Community Development Block Grant (CDBG) [4,7] and the Flood Mitigation Assistance Program, which also grant funds from the National Flood Insurance Program (NFIP) for property acquisition. Some of these funding agencies require a benefit-cost analysis process to determine whether it is more expensive to purchase the property rather than cover repairs and risk potential loss in the future. Damaged properties in the 100-year floodplain and those that have experienced severe repetitive losses are automatically considered cost effective for buyout and acquisition [7,9,10].

Although these mitigation methods are widely used by many local jurisdictions, among decision makers and planners, there is a discrepancy between where mitigation is implemented and an approach that is financially optimal from a flood-risk perspective. In addition, some approaches will protect more structures than other policies. This optimization issue is due in part uncertainties in data accuracy and limitations, as well as analyzing a limited number of potential policies. FEMA's Benefit-Cost Analysis (BCA) toolkit is widely used to justify property acquisition as economically viable [11], however it fails to account for data uncertainties. Failing to address uncertainties is not unique to flood risk reduction; it is also common across many forms of climate adaptation community plans [12]. These limitations result in mitigation measures that can be haphazardly administered following a flood event. A robust decision making (RDM) approach is one approach to making more informed decisions [13] and can be useful in identifying the optimal combination of buyout and elevation policies for flood prone communities. RDM methods are iterative, starting with a simple model, which gradually become more complex as more information is known [13]. RDM models identify slow moving variables, or constraining assumptions that can create undesirable outcomes [14].

The RDM process considers multiple objectives before a final decision. Through RDM, decision makers identify optimal outcomes given multiple objectives; however, the outcomes are often wide-ranging due to uncertainties in the decision-making parameters. Deep uncertainty occurs when the distribution of one or more variables is unknown and experts cannot agree [15]. However, with a properly designed model, an RDM provides scenarios and estimated outcomes, which are robust in accounting for uncertainty [16]. Uncertainty is explored through scenario-based models that iterate across the range for a variable. When a model uses a small sample of scenarios, they characterize the uncertainty, while a larger sample of scenarios allows for quantifying the uncertainty [17].

RDMs have a series of methods to quantify impacts of variable uncertainty. These methods focus on identifying either optimal scenarios or identifying conditions that create specific effects. When optimizing for multi-objectives, there are often a series of ideal solutions. Those scenarios that cannot increase the outcome for one objective without decreasing the outcomes of another are considered non-dominated and generate the Pareto front [18]. The Pareto front identifies the optimal scenarios given multi-objectives. Within two objective models, this front is graphically represented as a curve of scenarios. This curve can be identified either by using an optimization model, such as the Borg Multi-Objective Evolutionary Algorithm [19] or identified after numerous scenarios are run to visualize more possibilities [20]. The latter is computationally intensive but provides for further scenario analysis.

An alternative to identifying the Pareto front is to bound scenarios based on looser objectives and these characteristics are often based on catastrophic consequences [21], but they can also be used to identify more ideal characteristics [22]. A common method for identifying these scenarios is to use a Patient Rule Induction Method (PRIM) analysis [23]. The PRIM analysis identifies gradually decreasing boxes that fit scenarios of interest across one or more axes. Through this analysis rules are generated in which the scenarios fit, adding variables and constraining rules as the box becomes smaller. Three metrics identify how well these boxes fit the scenarios—(1) coverage explains how completely the objective is defined by the box, (2) density measures the purity of the scenarios and is analogous with precision, and (3) interpretability identifies how helpful this information is to decision makers [21]. Density and coverage are often diverging characteristics, therefore picking an appropriate box creates tradeoffs.

The current approach to buyout and elevation often uses limited analysis to identify an economically viable strategy to prevent future damages. Limited data on the influence of cost adjustments such as demolition cost, appraised market value of the property, structure elevation costs, and budgetary allocation are usually causes of deep uncertainty in decision making for urban planners and floodplain mangers. We apply an RDM approach to address these uncertainties and seek to answer the following research questions: What does the optimal allocation of flood mitigation resources look like under data uncertainty, and what policy practices will reduce undesirable scenarios?

## 2. Methods

We identify the optimal policies for buyouts and structural elevation using the Exploratory Modelling and Analysis (EMA) uncertainty workbench [24] and calculate BCRs for each parcel 10,000 times, using a Latin Hyper-Cube Sampling between ranges for each variable of interest. This sampling approach ensures that adequate scenarios are run. After the model is run, we identify the Pareto front. In addition, we use the full scenario set to identify rules for selected outcomes and use a PRIM analysis to determine what policies (i.e., structure selection and budget allocation) and uncertain variables will produce those effects (property values, construction and demolition costs). Below, we provide a full methodology on property selection.

### 2.1. Study Area

This study focuses on Galveston and Harris counties, located on the southeastern Texas coast in the US. The Houston Galveston Metropolitan Area (HGA) also includes Galveston Island, a barrier island in the Gulf of Mexico. The HGA is one of the largest metropolitan areas in the US, with approximately 4.42 million people, according to the 2010 US Census. Projections by the Houston Galveston Area Council estimates that the population of the area may surpass 9 million people by 2040 [25].

Proximity to the Gulf Coast, relatively flat topography, soil conditions, and climate, combined with the rapid population and economic growth, make the study area vulnerable to both coastal and inland flooding. The upper Texas coast is one of the most surge-prone regions in the US and on average experiences one major hurricane every 15 years [26]. Tropical storm Imelda is the most recent storm to hit the southwest coast of Texas in September 2019. Prior to that was Hurricane Harvey in

August 2017, with its extraordinary rainfall causing immense flooding and damage. Before Harvey, the surge associated with Hurricane Ike in 2008 resulted in unprecedented social and economic impacts locally [27]. Both Hurricane Ike and Hurricane Harvey were billion-dollar events, having estimated damages exceeding $35.1 billion USD and $125 billion USD, respectively [28].

## 2.2. Buyout and Elevation Analysis

Analysis for this study was conducted at the parcel scale. Between Galveston and Harris Counties we constructed a dataset consisting of 525,455 residential parcels that were provided by each county's tax assessors. Within our analysis, we first identify properties that would benefit from flood risk mitigation by selecting properties flooded by either Hurricane Harvey or Ike, where peak inundation from either storm is higher than the property's first floor elevation. Of those parcels eligible for a mitigation action, we calculated cost of acquisition or elevation, savings or damages prevented from a flood event, and the benefit to cost ratio (BCR) for both buyouts and elevation. We calculated buyout cost by adding the appraised market value of the property with the cost to demolish the property and restore it to open space. Next, we identified the total savings from buyouts as the total of average annualized losses (AAL) throughout the life of the property and added it to the present value of all flood claims for the property

The AALs represent a property's expected losses averaged for each year, these are estimated from AIR Worldwide Corporation, a pioneer in the catastrophe modeling industry [29]. Maintenance values were estimated at an annual cost of $550 USD per parcel based on conversations with community planners. Remaining Economic Life (REL) was used to estimate how long a structure will remain functional and was estimated using the parcels building quality and improvement type. Savings for each year from present to the end of the property's life is discounted by 3% as estimated by the US Office of Management and Budget, a rate used in previous buyout studies [9,30]. Finally, we calculate buyouts BCR by dividing buyout cost by buyout savings.

$$Buyout\ Cost = Market\ Value + Demolision\ Cost \tag{1}$$

$$Buyout\ Savings = \frac{\sum_{i=0}^{REL} AAL - Maintenance}{1.03^i} + flood\ damages \tag{2}$$

$$REL = Economic\ Life * (Improvement\ Value/Replacement\ Value) \tag{3}$$

Next, we calculate how much a structure should be elevated, as the depth of inundation from either Hurricane Harvey or Hurricane Ike depending on which was deeper for a structure. Elevation of a property was maxed out at five feet. This elevates the structure to the point where most would not be inundated by the selected storm event. Elevation cost was calculated by multiplying the cost per square foot to elevate, depending on foundation type, with the area of the building and the extra elevation to be added to the building. Initial estimates from local elevation companies in the Houston area for slab elevation is $65 USD per ft$^2$ and pile is $45 USD per ft$^2$. We estimated demolition costs for Elevation savings is estimated by subtracting the expected losses before elevation from expected losses after elevating the property. Expected property damage is estimated as property losses from either Hurricane Ike or Harvey minus first floor elevation using USACE depth damage curves (for further explanation of calculating inundation damages see [31]. Finally, BCR is calculated as elevation savings divided by elevation cost.

$$Elevation\ Cost = height\ elevated * cost_{sqfoot} * bldg\ area \tag{4}$$

$$Elevation\ Savings = property\ damage_{original} - property\ damage_{mitigated} \tag{5}$$

*2.3. Uncertainty Analysis*

To identify whether to elevate or buyout, we begin by selecting all parcels with BCR greater than 0.75, the minimum threshold value currently used when estimating the environmental benefits of property acquisition using FEMA's BCA toolkit [11]. We further sort the selected properties from highest to lowest BCR and apply a running sum of the cost. This allows us to subset parcels with an optimal BCR given a specific budget. To identify structures to elevate, the same approach is used for those parcels not meeting the buyout criteria for elevation. As shown in Table 1, the uncertainty analysis allows policy makers to identify scenarios that allow for budgets allocation of up to $750 million USD, buyout proportions 10–90%, as well as options for elevation cost, demolition cost, and appraised market value of the elevated or acquired property.

**Table 1.** Summary of policy and uncertainty options.

| Analysis Category | Variable | Value Ranges |
|---|---|---|
| Policies | Budget<br>Proportion buyouts | $75–750 million USD<br>10–90% |
| Uncertainties | Demolition cost<br>Appraised market value cost adjuster<br>Elevation cost adjuster | $6–9 USD per foot $^2$<br>90–130%<br>70–130% |

Our uncertainty analysis yielded a total of 10,000 scenarios with the following set of outcomes:

- Buyout structures: the number of recommended structures to be bought out given uncertainties;
- Elevation structures: the number of recommended structures to be elevated given UNCERTAINTIES;
- Buyout savings: total savings from bought-out structures given uncertainties;
- Buyout cost: total cost from bought-out structures given uncertainties;
- Buyout BCR: total benefit to cost ratio from bought-out structures given uncertainties;
- Elevation savings: total savings from elevated structures given uncertainties;
- Elevation cost: total cost from elevated structures given uncertainties;
- Elevation BCR: total benefit to cost ratio from elevated structures given uncertainties.

## 3. Results

*3.1. Uncertainty Scenarios*

Results show that, on average, more properties are proposed to be impacted through elevation than buyouts, with a higher return on investment. When mitigation strategies are combined, overall BCRs are reduced compared to a single elevation or buyout option. As shown in Table 2, our analysis created 10,000 potential scenarios which accounted for variations in the budget, proportion spent on buyouts, and uncertainties in demolition cost, property market values, and cost to elevate a structure. Across these scenarios, return on investment averaged $2.45 USD for each dollar spent and would on average protect 2998 properties when both buyouts and elevation strategies are combined. Elevating structures would often benefit more properties, with an average of 1672 properties protected from inundation with a mean BCR of 2.45 while BCRs were often slightly higher with buyouts but protecting fewer properties from flooding.

**Table 2.** Descriptive statistics of the outputs for the 10,000 scenarios given uncertainties and policy variations.

| Category | Variable | All Scenarios | | | |
| --- | --- | --- | --- | --- | --- |
| | | Mean | STD | Min | Max |
| Buyouts | Number of structures | 1326.64 | 847.15 | 79 | 4520 |
| | Savings ($ million USD) | 494.8 | 244.2 | 55.8 | 1272.6 |
| | Cost ($ million USD) | 206.0 | 1.43 | 8.45 | 669.7 |
| | BCR | 2.88 | 0.835 | 1.55 | 7.73 |
| Elevation | Number of structures | 1672.14 | 1034 | 854 | 6143 |
| | Savings ($ million USD) | 428.5 | 235.4 | 45.7 | 1374.9 |
| | Cost ($ million USD) | 206.2 | 143.1 | 7.5 | 659.3 |
| | BCR | 2.45 | 0.784 | 1.11 | 8.09 |
| Both | Number of structures | 2998.78 | 1296.99 | 622 | 6752 |
| | Savings ($ million USD) | 923.3 | 330.9 | 230.5 | 1747.8 |
| | Cost ($ million USD) | 412.2 | 194.9 | 74.8 | 749.7 |
| | BCR | 2.45 | 0.515 | 1.43 | 4.49 |

*3.2. Pareto Front Analysis*

To achieve our objective of maximizing economic benefits as well as removing the maximum number of structures out of flooding, we perform a Pareto front analysis. Maximizing both BCRs and number of structures protected are diverging objectives, and graphing the Pareto front provides a clear understanding of how these objectives are related. Higher budgets will increase the number of structures protected; however, it decreases the amount saved per dollar spent. Calculating the Pareto front for these 10,000 scenarios identified 32 scenarios (see Figure 1) that performed better across BCR and the number of structures. We identified three scenarios that fall along this Pareto front—the first prioritizes BCR, and the second prioritizes structures. The last scenario represents the middle ground, where neither objective is at their maximum (Table 3). Along this front, the maximum BCR was 4.49 which would protect 1065 properties. Maximizing structures would protect 6752 properties, but the BCR is only 2.22.

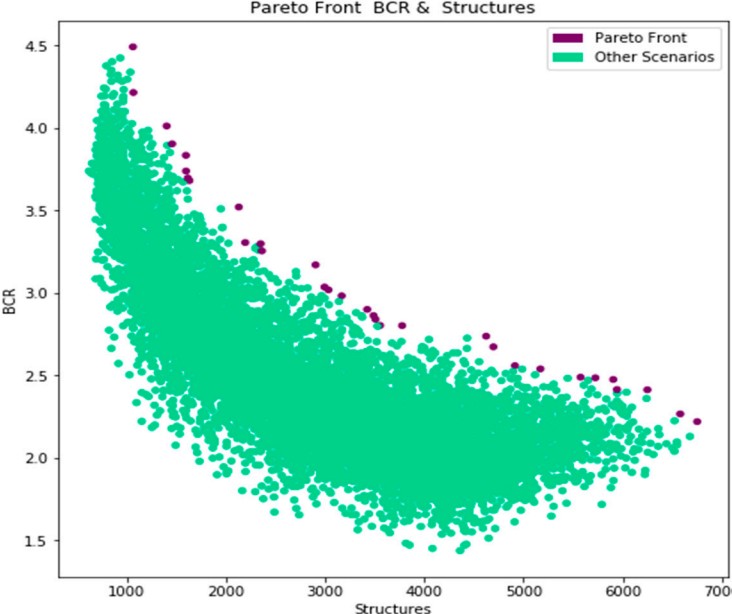

**Figure 1.** Pareto front for benefit to cost ratio (BCR) and number of structures.

**Table 3.** Summary of three Pareto front experiments.

| Category | Variable | Combined BCR & Structures | Maximize BCR | Maximize Structures |
|---|---|---|---|---|
| Buyouts | Number of structures | 234 | 184 | 609 |
| | Savings ($ million USD) | 131.4 | 108.5 | 282.1 |
| | Cost ($ million USD) | 30.8 | 23 | 89 |
| | BCR | 4.26 | 4.71 | 3.17 |
| Elevation | Number of structures | 1898 | 881 | 6143 |
| | Savings ($ million USD) | 574.1 | 293.2 | 1375 |
| | Cost ($ million USD) | 169.6 | 66.4 | 657 |
| | BCR | 3.83 | 4.41 | 2.09 |
| Both | Number of structures | 2132 | 1065 | 6752 |
| | Savings ($ million USD) | 705.5 | 401.7 | 1657 |
| | Cost ($ million USD) | 200.4 | 89.4 | 746 |
| | BCR | 3.52 | 4.49 | 2.22 |

Figure 1 shows the Pareto front compared with all other scenarios. This front was identified by maximizing both the BCR and numbers of structures. Each point represents a scenario closest to optimal (maximum number of structures and highest BCR) where any further increase in either BCR or number of structures, will subsequently decrease the other.

The results for the Pareto front are shown in Tables 3 and 4. Table 3 shows the distribution of the number of structures, cost, benefits and BCRs for maximizing either number of structures, BCR, or both. Table 4 shows the policy and uncertainty values for these three possible scenarios along the Pareto front. As shown in Tables 3 and 4, in order to maximize the return on investments only, it will require a lower budget of about $90 million USD with a primary focus on spending about 74% of the budget on elevation. The experiment to maximize number of protected structures increases the budget to about $700 million USD, and the policy should be further dominated by elevated structures with about 88% of the budget. Across all three scenarios, actual market values should be higher than appraised market values, while elevation costs should be cheaper. Demolition costs are slightly more expensive than the average scenario ($7.55–8.29 USD per square foot).

**Table 4.** Policy Uncertainty values across 3 scenarios.

| | Budget ($ Million USD) | Proportion of Budget for Buyouts (%) | Demolition Cost ($ USD Per Foot $^2$) | Elevation Cost (%) | Market Value (%) |
|---|---|---|---|---|---|
| Combined BCR and structures | 200.48 | 14.4 | 8.29 | 70.06 | 128.11 |
| Maximize BCR | 90.02 | 26.19 | 7.69 | 70.81 | 128.44 |
| Maximize structures | 746.52 | 11.95 | 7.55 | 72.34 | 117.55 |

*3.3. Spatial Distribution of Scenarios*

Figures 2–4 show the spatial distribution of the three Pareto front experiments and identifies properties that are either to be elevated or bought out. These proposed scenarios are compared with Post-Harvey buyout focused areas within Harris County. Figure 3 specifically shows the distribution of structures that equally prioritize both return on investment and number of structures. The results show that most of the properties recommended for elevation are around the coastal communities of Galveston with a few clusters around Galveston Island that are recommended for buyouts due to their location and exposure to repetitive flooding. The areas recommended for buyouts in this research are well in line with buyout-focus areas identified by Harris county agencies after Hurricane Harvey (see reference [32]). However, our analysis shows that there are still several areas where buyouts will be cost effective that are not currently identified as buyout areas in Harris County. Additionally, several

clusters of properties that our analysis recommended for elevation are still identified by Harris County agencies as buyouts using Hurricane Harvey buyout funding.

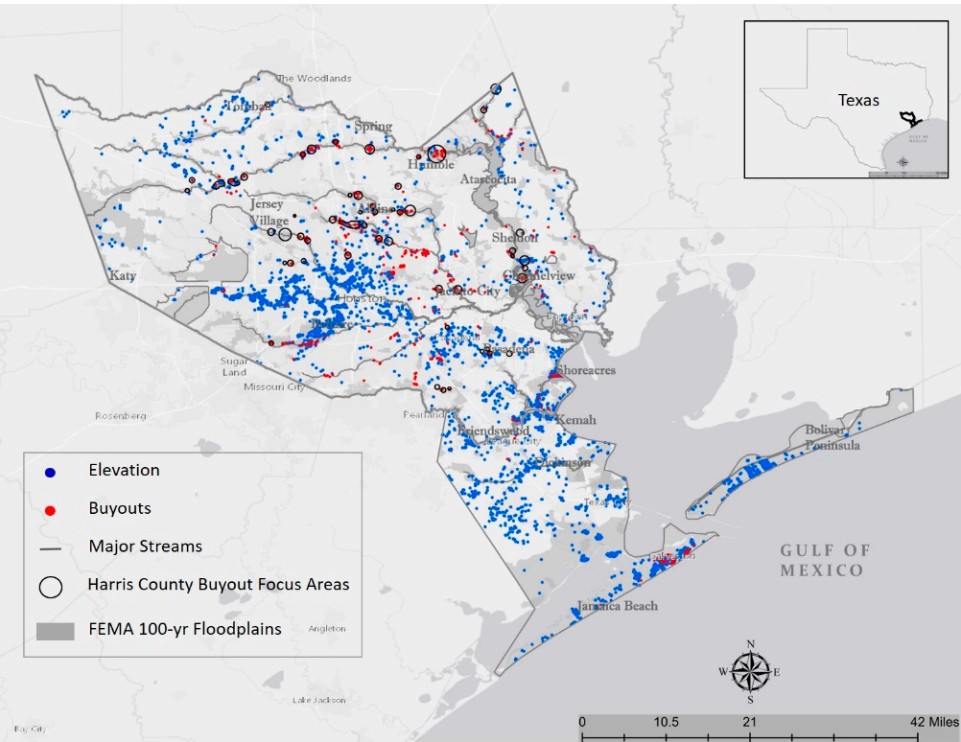

**Figure 2.** Spatial locations of optimizing both BCR and number of structures with post-Harvey buyout–focused areas.

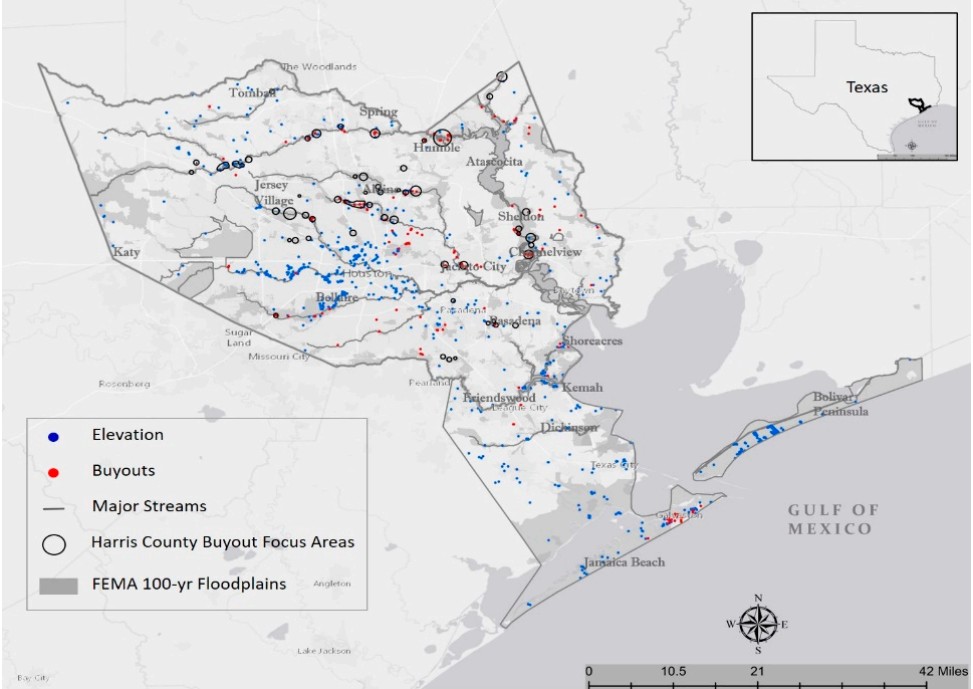

**Figure 3.** Spatial locations of optimizing BCR with post-Harvey buyout focused–areas.

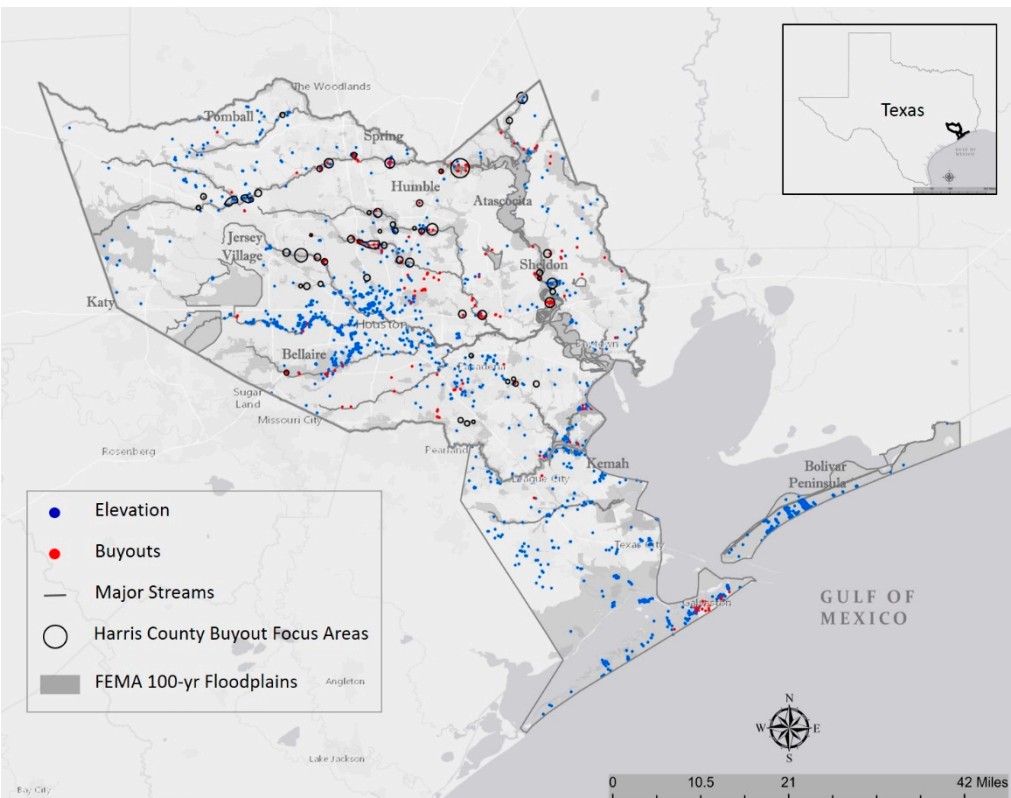

**Figure 4.** Spatial locations of optimizing both Structure and BCR with post-Harvey buyout–focused areas.

For maximizing the number of structures, Figure 4 shows a similar trend in terms of the locations of buyouts and elevation, but with a higher overall number of structures that are mitigated from flood damages. The results also show that several structures outside the SFHA are selected for elevation especially around central Houston, Bellaire and downstream of the Addicks and Barker Reservoirs, highlighting the need to address flood risk for properties outside the FEMA-designated 100-year floodplain as economically viable properties for flood hazard mitigation. This scenario also captures several properties in the coastal communities of Galveston and Bolivar Peninsular as prime properties for additional elevation to reduce flood risk.

Figure 5 shows the spatial distribution of a policy focused on maximizing the BCR, which also leads to several structures recommended for elevation. Although fewer structures are recommended for buyout and elevation, this strategy has the highest return on investment compared with other scenarios, with a BCR of almost 4.5. The spatial distribution highlights the locations of the properties that either need buyouts or elevation that would provide the most value for every dollar spent in the study area. The pattern shows that properties that are in the SFHA or in proximity to streams and bayous are prioritized. Similarly, the two additional scenarios reviewed identified clustered areas that have a higher return on investment if elevated, whereas local officials are targeting these areas for buyouts.

*3.4. PRIM Results*

The PRIM analysis optimized two criteria—an ideal scenario should have a BCR of at least 2.19 and protect at least 2350 structures (Figure 5). Through this analysis, three rules were identified for scenarios to meet the objectives: (1) The budget should be between $301.9–531.9 million USD; (2) Elevation costs should be between 70% and 102% of FEMA estimated costs; (3) The proportion of the budget focused on buyouts should be between 17–85%. Generating a density of 90.4% and a coverage of 51.4%, these policies will have a 90% chance of achieving an optimal outcome, while 49.6% of optimal scenarios fall outside these condtions. These conditions will hold up across the data

uncertainties of market value and demolition costs; however, elevation costs need to be close to or cheaper than estimated costs. To ensure that elevation costs are not a factor, the budget and proportion of buyouts will have to be further refined. Budget needs to be \$340–435 million USD, while the ratio should be 36.4–87.2% for buyouts. This scenario keeps density near the same (90.2%), but coverage is dropped to 29.1%, suggesting there are numerous viable options outside these scenarios.

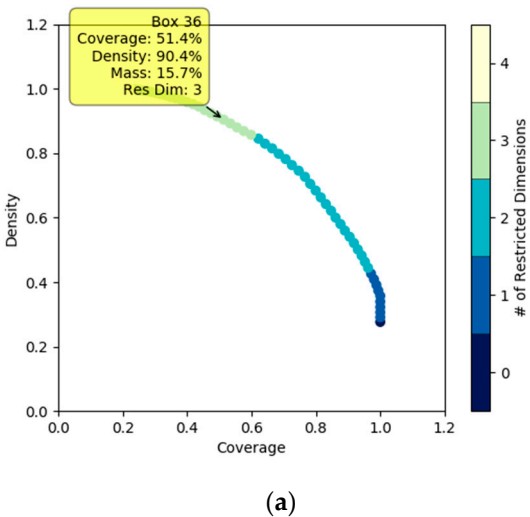

(**a**)

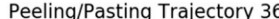

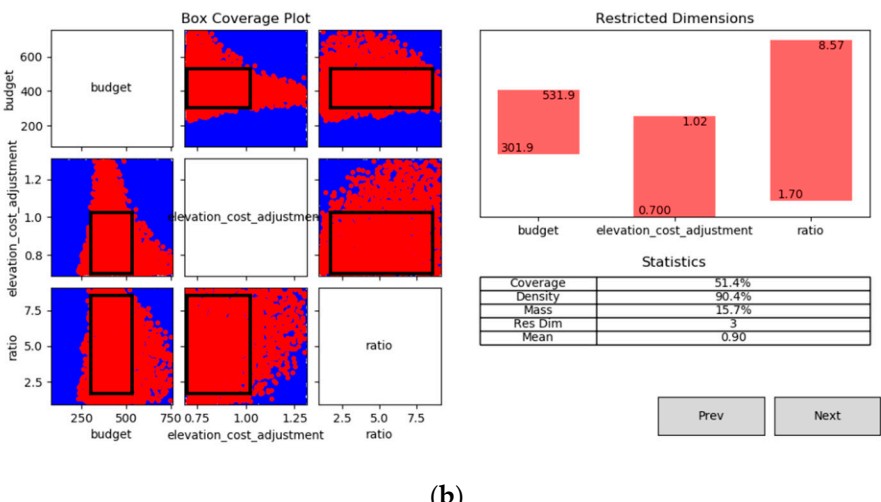

(**b**)

**Figure 5.** Patient Rule Induction Method (PRIM) analysis for optimal scenarios. (**a**) shows the PRIM analysis for the objective: BCR > 2.19 and Structures > 2350. This represents 0.5 standard deviation below each mean. (**b**) represents the rules generated by box 36 and will result in Coverage = 51.4% Density 90.4%.

## 4. Discussion

Flood exposure and risk are expected to increase due to rapid population and development pressure in vulnerable locations, while uncertain futures require a more robust process [12]. Our research contributes to a growing need for evaluating flood mitigation strategies [33] while also accounting for data uncertainties as a more robust approach. Although development policies may be in place to prevent new development in floodplains and other vulnerable locations, it is essential that existing properties are also either protected from flooding or completely removed from flood prone areas to prevent the continued economic burden of repetitive flood losses. Our research has proposed a robust approach to estimating possible outcomes for differing policies for elevating and buying out

properties. These findings highlight the importance of evaluating parcel level mitigation strategies [34]. Our scenarios maximize number of properties protected for the most economic benefit and would expect to experience reduced or no flood damages from future flood events. In comparison with the status quo [32], this robust approach identifies a larger pool of potential properties to be mitigated. These set of results provide local jurisdictions an opportunity to select suitable mitigation strategies that are in line with their development policies, while also providing economically viable mitigation options to homeowners. Below we highlight some important findings of this analysis.

Our analysis provides policy-based scenarios that are robust against several uncertainties. Most decision makers ignore the uncertainties in the cost and budgetary allocations for mitigation projects and make ad hoc, haphazard mitigation practices that are usually implemented in a reactionary manner to specific flood event thereby resulting in minimal impacts [2]. Addressing the uncertainties associated with mitigation measures not only provides an objective measure of identifying mitigation options, but also ensure the appropriate mitigation measure given budget, locational and uncertainty constraints. Money for hazard mitigation budgets are often allocated by the federal government and are expected to be spent with utmost care and provide high return on investment [3]. Our results show that budget drives most of the variations in outcomes when looking at elevation and buyouts combined. Higher budgets would protect more properties, but with an associated decrease in economic efficiency, a result demonstrated in previous multi-objective mitigation studies [35]. When looking at buyouts and elevation individually, the proportion allocated to buyouts had a higher impact in variation and outcomes.

The results of our experiments suggest that decision makers have the ability to achieve high BCR values while protecting thousands of structures, even when facing uncertainties. Uncertainties in market value were the largest driving factor relative to uncertainties around demolition and elevation costs. Scenarios were more likely to achieve our objectives, if the appraised market value used in the study are over-valued compared with actual market values. When market values are lower, as compared to appraised property values, more structures can be protected or removed, while a higher market values increases the amount saved per dollar spent. This highlights the importance of conducting additional research on when and how properties should be appraised for government acquisition to prevent flood damage. Further, while other studies suggest that elevation will reduce damages, without grants to help homeowners, retrofitting homes to a higher elevation may be cost prohibitive for many structures [36]. This study provides evidence that elevating structures should be a more viable option to spend government resources for risk mitigation when the appropriate funding mechanisms are in place, as our analysis showed that elevation consistently protected more properties across scenarios while remaining cost efficient.

Although this study explored mitigation alternatives using multiple-objectives and inclusive of uncertain parameters, this model was a simple first step and future research should be conducted to address our limitations. For example, we focused on parcels within two counties, but most of the decisions will be made at the municipal level. In Harris County alone, there are 45 different incorporated municipalities, each of which has their own tax base, development policies, and capacity to support flood mitigation projects. While municipalities provide mitigation options such as elevation or buyouts, the ultimate decision lies with the homeowner. Because some homeowners may push back against governmental property acquisition, a checkerboard pattern often occurs where buyouts are not directly adjacent to other buyouts or existing open space. This is a major critique of buyout programs because it limits the ecological benefits of open space restoration [37]. Future research should identify policies that incentivizes clustered buyouts as part of the process and account for current homeowner incentives. For example, the Harris County buyout program provides incentives such as relocation fees and prioritizes socially vulnerable households to encourage participation [38].

In addition, our approach makes other assumptions that could be explored as uncertainties, such as discount rates and AALs. In addition, this model was static in time. Temporal uncertainties in the region such as increased built environment, sea level rise, and climate change, could adversely impact

the region. A more robust approach that account for these temporal variations can effectively identify economically viable parcels for mitigation before their risk is increased. Finally, the model uses two extreme storms to identify flood impact and elevation. Future research should explore more events and better quantify risk to identify potential mitigation parcels.

## 5. Conclusions

Our analysis provides a robust method of decision making in the flood mitigation and planning process. The results were measured using multiple objectives through a Pareto front to generate robust projections of BCRs and properties protected by either elevation or property acquisition. The findings accounted for three data uncertainties including market value, the cost to elevate a property, and the cost to demolish a structure. These uncertainties and policy variations that include changing budgets and project allocations was simulated through 10,000 scenarios in the Houston-Galveston Region. This approach provides planners and policy makers with a viable method to implement both rational and incremental planning that accounts for uncertainties and assumptions to make more informed decisions. Additionally, the PRIM analysis found an ideal set of policy constraints can help to boost both property selection and increase the overall return on investment for the selected mitigation options. Our findings suggest that a robust decision-making process will benefit both vulnerable individuals and the Federal Government through reduced disbursement of disaster grants for mitigation. Finally, our research helps local jurisdictions identify areas that need protection which have been missed, or those that have the wrong form of flood mitigation proposed.

**Author Contributions:** Conceptualization, W.M., K.O.A., and W.E.H.; methodology, W.M. and K.O.A.; software, W.M.; validation, K.O.A. and W.E.H.; formal analysis, W.M.; investigation, W.M. and K.O.A.; resources, W.E.H.; data curation, K.O.A.; writing—original draft preparation, W.M. and K.O.A.; writing-review and editing, W.M., K.O.A., and W.E.H.; visualization, W.M. and K.O.A.; supervision, W.E.H.; project administration, W.E.H.; funding acquisition, W.E.H. All authors have read and agreed to the published version of the manuscript.

**Funding:** This research was funded by USA National Science Foundation Grant No. 1545837 and Texas General Land Office Contract No. 19-181-000-B574.

**Acknowledgments:** The authors would like to acknowledge the Sustainable Climate Risk Management group from Penn State for their expertise and advice.

**Conflicts of Interest:** The authors declare no conflict of interest

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
