# Peer review of "Uncertainty in Flood Mitigation Practices: Assessing the Economic Benefits of Property Acquisition and Elevation in Flood-Prone Communities"

_sustainability, doi:10.3390/su12052098_

Round 1

Reviewer 1 Report

The authors seek to identify whether property elevation or buyouts are the most economically beneficial option for the flood-prone counties of Harris and Galveston, Texas using a robust decision-making approach. This article is timely and incredibly relevant to the current mitigation issues facing communities; it is actually overdue, considering the 40-plus year history of buyouts in the U.S. and no substantial empirical studies evaluating the cost-benefit of them. The methodology is novel and clearly explained. The conclusions seem appropriate to the results of the study.

I particularly like the mitigation comparison between BFE and buyouts as communities and residents have options when deciding how to mitigate. The authors could strengthen their argument even more by highlighting this choice, or even acknowledging the residential choice in the mitigation process. Buyouts, as funded by FEMA, HUD, and other federal agencies, are voluntary and this type of cost-benefit analysis may be a useful tool for communities to help homeowners select the appropriate mitigation technique for themselves. I do think the manuscript would benefit from acknowledgment of the role of homeowners in the mitigation decision-making; as it’s currently presented, the manuscript only discusses municipalities selecting buyout zones.

Additionally, the authors mention the calculated buyout costs they used (lines 144-145), but do not mention the associated relocation fees (Harris County currently pays them on top of the home value) nor open space management. In the introduction, they state that the land goes back to a natural state, but the literature shows most buyout land is mowed or hosts some use beside reverted back to unmaintained land with no associated costs (Zavar's work). One limitation to acknowledge is the on-going cost of post-buyout land management. Communities spend sizable amounts on the mowing and upkeep of these open spaces, Harris County alone spends millions each year. Relatedly, I wonder if the scenarios consider checkerboarding; if adjacent homeowners all participate in the buyout, the maintenance costs go down, thus impacting this CBR. I’m not suggesting these be rerun, but rather a discussion point to acknowledge and potential future work.

In the discussion section (lines 303-315), the authors can push their argument more by drawing links to the literature on the socio-economic implications of selecting homes with lower market values for buyouts. There’s a growing body of work (Loughran and Elliott’s work in Houston) discussing social inequities related to buyouts. 

General Comments:

Abstract: Identify that you are using RDM approach in your abstract and key words as it may make your article more discoverable to those interested in the method.

I appreciate the inclusion of the spatial distribution, maps, and comparison of the results to existing buyout priority zones. The maps are easy to read. I would clarify that the Harris County Buyout Focus Areas are the post-Harvey locations as they’ve been running buyouts for decades. I wonder how the previous buyout areas compare to your map locations; a topic for future study, perhaps.

In the discussion section, the reference for future work (line 322) on where residents could potentially move once bought out feels out of place and ignores the robust body of lit coming primarily from Brokopp Binder, McGee, and Greer on the topic.

 Please edit carefully; there’s several typos throughout including missing commas, needed semicolons (e.g. line 266), and a missing period (e.g. line 147).

Author Response

We would like to thank the anonymous reviewer for the thorough response to our paper. Many of the comments and suggestions have led to insightful discussions and prompted us to improve clarity of the text in the paper. We believe that the comments provided by the reviewers and the changes we have made have significantly improved our paper, and we hope that the reviewers will agree. 

Reviewer 1.  

1. I particularly like the mitigation comparison between BFE and buyouts as communities and residents have options when deciding how to mitigate. The authors could strengthen their argument even more by highlighting this choice, or even acknowledging the residential choice in the mitigation process.  

In lines 289-291, the authors further highlight this comparison. We mention how jurisdictions have an option of channeling different mitigation techniques to complement their development and land use policies while also providing homeowners mitigation options that are economically viable. 

2. I do think the manuscript would benefit from acknowledgment of the role of homeowners in the mitigation decision-making; as it’s currently presented, the manuscript only discusses municipalities selecting buyout zones. 

3. Additionally, the authors mention the calculated buyout costs they used (lines 144-145), but do not mention the associated relocation fees (Harris County currently pays them on top of the home value) nor open space management. In the introduction, they state that the land goes back to a natural state, but the literature shows most buyout land is mowed or hosts some use beside reverted back to unmaintained land with no associated costs (Zavar's work). Relatedly, I wonder if the scenarios consider checkerboarding; if adjacent homeowners all participate in the buyout, the maintenance costs go down, thus impacting this BCR. I’m not suggesting these be rerun, but rather a discussion point to acknowledge and potential future work. 

The reviewer raises a good point about relocation and management fees. We also further highlight the importance of providing incentives for participation and cite Harris county as an example in lines 323-331. We also discuss the role emphasize that while jurisdictions provide mitigation options to the community, it is ultimately the homeowner’s decision to choose which mitigation strategy to adopt. 

4. One limitation to acknowledge is the on-going cost of post-buyout land management. Communities spend sizable amounts on the mowing and upkeep of these open spaces, Harris County alone spends millions each year.  

On line 150 we address the maintenance costs for mowed land.

5.  In the discussion section (lines 303-315), the authors can push their argument more by drawing links to the literature on the socio-economic implications of selecting homes with lower market values for buyouts. There’s a growing body of work (Loughran and Elliott’s work in Houston) discussing social inequities related to buyouts.  

While socio-economic implications of buyouts are important to address, in the paper we are assessing data issues and systemic undervaluing of properties. We think there was some confusion in the way we addressed this. Throughout the paper, we now refer to market value as ‘appraised market value’, what we get from the county appraisal district and ‘actual market’ which is modified by our analysis. We define the terms ‘market value’ and ‘appraised market value’ earlier in the paper as the uncertainties associated with cost adjustments. 

Further the discussion regarding market value is changed to the following: 

“Scenarios were more likely to achieve our objectives, if the appraised property value used in the study are over-valued compared with actual market values. When market values are lower, as compared to appraised property values, more structures can be protected or removed, while a higher market values increases the amount saved per dollar spent.” 

General Comments:

6. Abstract: Identify that you are using RDM approach in your abstract and key words as it may make your article more discoverable to those interested in the method. 

We added “Using a Robust Decision Making (RDM) approach, the effects of uncertainty in property values...” to the abstract.  

7. The maps are easy to read. I would clarify that the Harris County Buyout Focus Areas are the post-Harvey locations as they’ve been running buyouts for decades.  I wonder how the previous buyout areas compare to your map locations; a topic for future study, perhaps. 

The Following sentence was added at lines 235, 236 “These proposed scenarios are compared with Post-Harvey buyout focused areas within Harris County.” Figure captions were also updated to reflect this change.  

 Previous buyouts would be an interesting, however, since they’ve already been bought out they would not be identified through this process and comparing the two would be difficult. 

8. In the discussion section, the reference for future work (line 322) on where residents could potentially move once bought out feels out of place and ignores the robust body of lit coming primarily from Brokopp Binder, McGee, and Greer on the topic. 

After discussion we agree that this sentence was out of place and was removed.  

9. Please edit carefully; there’s several typos throughout including missing commas, needed semicolons (e.g. line 266), and a missing period (e.g. line 147). 

 The Authors edited the document again for grammatical errors. 

Reviewer 2 Report

Dear Authors,

The novelty and structuring of the paper are good. Please update the funding and acknowledgment sections. Can you please explain more about Pareto front analysis in the manuscript. 

Thanks!

Author Response

We would like to thank the anonymous reviewer for the response to our paper. Many of the comments and suggestions have led to insightful discussions and prompted us to improve clarity of the text in the paper. We believe that the comments provided by the reviewers and the changes we have made have significantly improved our paper, and we hope that the reviewers will agree. 

1) Please update the funding and acknowledgment sections.  

These have been updated.  

2) Can you please explain more about Pareto front analysis in the manuscript. 

The following sentence was added on line 93. “The Pareto front identifies the optimal scenarios given multi-objectives.”